# Action sharpens sensory representations of expected outcomes

Daniel Yon[1], Sam J. Gilbert [2], Floris P. de Lange[3] & Clare Press[1]

When we produce actions we predict their likely consequences. Dominant models of action control suggest that these predictions are used to 'cancel' perceptual processing of expected outcomes. However, normative Bayesian models of sensory cognition developed outside of action propose that rather than being cancelled, expected sensory signals are represented with greater fidelity (sharpened). Here, we distinguished between these models in an fMRI experiment where participants executed hand actions (index vs little finger movement) while observing movements of an avatar hand. Consistent with the sharpening account, visual representations of hand movements (index vs little finger) could be read out more accurately when they were congruent with action and these decoding enhancements were accompanied by suppressed activity in voxels tuned away from, not towards, the expected stimulus. Therefore, inconsistent with dominant action control models, these data show that sensorimotor prediction sharpens expected sensory representations, facilitating veridical perception of action outcomes.

[1] Department of Psychological Sciences, Birkbeck, University of London, London WC1E 7HX, UK. [2] Institute of Cognitive Neuroscience, University College London, London WC1N 3AZ, UK. [3] Donders Institute for Brain, Cognition and Behaviour, Radboud University, Nijmegen 6525 HR, Netherlands. Correspondence and requests for materials should be addressed to D.Y. (email: d.yon@bbk.ac.uk)

Action is the only means we have of influencing the world around us. It has been appreciated for over a century that effective action depends on predicting its sensory consequences[1]. We select actions based upon their predicted outcomes[2] and can use these predictions to generate rapid corrective movements when we experience deviant sensory input[3]. Influential models of action control developed in recent decades have proposed that these predictions are used to suppress, or 'cancel', perceptual processing of expected action outcomes. Namely, it is assumed that a 'forward model' in the motor system suppresses activity in expected sensory units[4], which allows agents to ignore predictable sensations and therefore remain maximally sensitive to unexpected outcomes that may be important for learning or planning new actions[3,5]. Such a cancellation mechanism may provide an explanation for why it is difficult to tickle oneself[6] and is thought to play a key role in generating our sense of agency and explain its aberration in psychiatric illness[7]. The idea has drawn wide support from neuroimaging studies that report predictable tactile[5,8] and visual[9–11] consequences of action are associated with reduced activity in sensory brain regions.

However, the core tenet of cancellation models—that sensory processing of predicted action inputs is suppressed—contrasts dramatically with predictive processing models developed in the wider sensory cognition literature. Bayesian accounts of perception typically emphasise that in an inherently noisy sensory world it is adaptive for observers to incorporate their prior expectations into their sensory estimates[12]. Mechanistically, this incorporation is implemented by altering the weights on sensory channels and effectively 'turning up the volume' on expected relative to unexpected inputs[13] (see Fig. 1). These 'sharpening' models are thought to explain a range of perceptual phenomena whereby observers are biased towards perceiving stimuli that they expect, for instance, perceiving greyscale bananas to be yellow[14]. Under these accounts it is hypothesised that activity in sensory brain areas may in principle be suppressed for expected inputs[15,16], but the suppression would not resemble that predicted by the cancellation account. Specifically, activity should be suppressed only in units tuned away from expected inputs, rather than in units tuned towards these inputs as hypothesised by the cancellation account. In line with this account, Kok et al[17]. found that visual stimuli that were expected on the basis of a preceding tone evoked weaker responses in primary visual cortex (V1) primarily in voxels tuned away from the presented stimulus, and multivariate pattern analysis (MVPA) demonstrated a superior ability to decode the observed stimulus from activity patterns in this area. Such findings suggest that weaker patterns of univariate activity can reflect a 'sharpening' of neural populations toward expected sensory events, rather than a suppression of expected signals.

To date, similar analysis techniques have not been applied to sensory signals predicted by action. As such, it remains unclear whether sensorimotor predictions act to suppress sensory activity associated with expected action outcomes, as is widely assumed in the action literature, or instead to sharpen such representations. We adjudicated between these possibilities by requiring human participants to execute hand actions (finger abductions) and simultaneously observe congruent (same finger) or incongruent (different finger) movements of an avatar hand, while recording neural activity using functional magnetic resonance imaging (fMRI). This congruency manipulation exploits the fact that congruent action outcomes are more expected than incongruent ones, based either on inherited evolutionary expectations or our extensive experience of controlling our own actions[2]. The attentional relevance of observed movements was orthogonally manipulated by alternating the participants' task between blocks.

Under the cancellation account, suppression of units tuned to the expected stimulus would reduce the amount of information about observed hand movements, impairing classifier performance on congruent trials and generating reduced activation in units tuned to the congruent stimulus (see Fig. 1). Conversely, if prediction sharpens populations towards expected outcomes, population responses will contain more information about observed hand movements on congruent trials and there will be reduced activation in units tuned away from the congruent stimulus. We find the latter pattern across early and late visual brain areas, suggesting that the predictions we make on the basis of action in fact sharpen sensory representations, facilitating veridical perception of our actions in an inherently uncertain sensory world.

## Results

**Manipulating expectations during action**. Twenty healthy human participants were instructed to abduct their index or little finger on the basis of a shape cue (see Fig. 2a). An observed avatar hand performed an abduction movement that was congruent or incongruent with the participants' own action and synchronous with it. The attentional relevance of the avatar movements was manipulated between blocks by requiring participants to judge a property of the observed movement (finger-judgements; e.g., 'Did the INDEX finger move?') or a coloured dot also presented on the display (colour-judgements; e.g., 'Was the dot BLUE?'). Analysis of participant accuracies confirmed that participants had little difficulty in following task instructions (finger-judgement trials: mean accuracy = 91.4 %, SEM = 0.022%; colour-judgement trials: mean accuracy = 95.6%, SEM = 0.018%; see Methods for further analysis of the behavioural data). There were also no move trials, used to define regions of interest, where participants simply observed the movements without performing actions.

**Enhanced decoding of congruent observed actions**. Linear support vector machines (SVMs) classified the identity of the observed action stimulus (index finger movement vs little finger movement) from patterns of neural activity (see Fig. 2b). 'Searchlight' analyses[18] yielded decoding maps for each participant. Decoding maps from no move trials were used to generate data-driven regions of interest for subsequent analysis[19]. A group-level $t$-test applied to these maps revealed three clusters across occipital and temporal regions where stimulus identity could be reliably decoded above chance (voxel-wise $p < 0.001$, cluster-wise FWE $p < 0.05$;[19,20]): bilateral occipital cortex, left occipital cortex and right occipitotemporal cortex.

To evaluate how predictions during action influence the quality of underlying sensory representations, we separately extracted decoding accuracies from these clusters for trials where the observed outcome was congruent or incongruent with the participant's own action (see Fig. 3). Comparison of these accuracies revealed an effect of congruency ($F_{1,19} = 4.781$, $p = 0.041$, $\eta_p^2 = 0.201$) that did not change across clusters ($F_{2,38} = 0.276$, $p = 0.760$, $\eta_p^2 = 0.014$), and which reflected superior decoding of stimulus identity on congruent trials relative to incongruent trials. Contrary to cancellation models, this result is consistent with the idea that prediction during action enhances the quality of sensory representations associated with expected outcomes.

To investigate possible interactions between effects of expectation and attentional relevance, we performed the same analysis separating finger-judgement and colour-judgement trials. This analysis found no interaction between congruency and task ($F_{1,19} = 3.42$, $p = 0.08$, $\eta_p^2 = 0.153$), but an interaction between congruency, task and cluster ($F_{2,38} = 3.99$, $p = 0.027$, $\eta_p^2 = 0.174$). Post-hoc analyses found this effect to reflect an interaction only in the right occipitotemporal cluster ($F_{1,19} = 8.122$, $p = 0.010$.

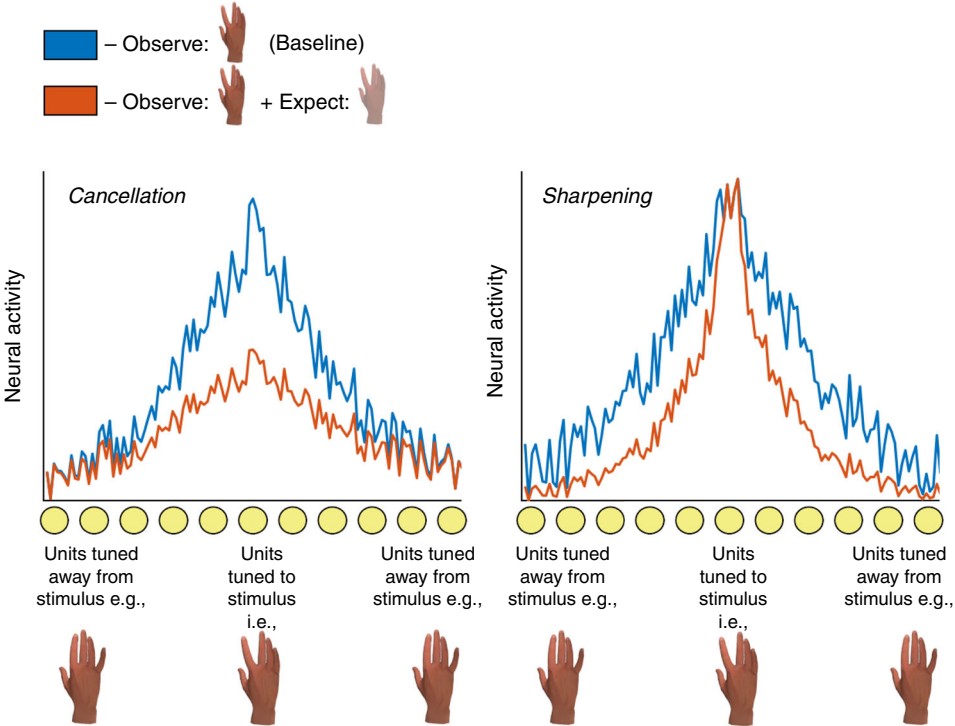

**Fig. 1** A schematic illustration of how predictive signals influence populations of sensory neurons under 'cancellation' and 'sharpening' models. Left: Cancellation models developed in the action control literature propose that when we move (e.g., abduct our index finger) we generate a predictive signal that suppresses activity in sensory units tuned to the expected action outcome (e.g., observation of an index finger movement)[3, 4]. Weakening the activity in these units reduces the signal-to-noise ratio of the population response when stimuli are expected, leaving less information in patterns of neural activity that could be detected by a pattern classifier. Right: In contrast, sharpening models of prediction found in the wider sensory cognition literature suggest that predictive signals suppress activity in units tuned away from expected stimuli[13, 17]. This leads to a sharper population response when stimuli are expected, with a higher signal-to-noise ratio and making it easier to decode stimulus identity on these trials

$\eta_p^2 = 0.299$; i.e., the smallest cluster), driven by stronger congruent enhancements when stimuli were task irrelevant. The congruency effect did not interact with task relevance in the bilateral occipital ($F_{1,19} = 1.705$, $p = 0.207$. $\eta_p^2 = 0.082$) or left occipital clusters ($F_{1,19} = 0.023$, $p = 0.880$. $\eta_p^2 = 0.001$), suggesting that the attentional relevance of the stimuli did not mediate the influence of expectation on sensory processing.

**Suppression in units tuned away from expected stimuli.** Superior decoding of stimulus identity on congruent trials could be achieved by relatively suppressing activity in units tuned away from the expected stimulus, yielding sparser population codes that are easier to distinguish with a pattern classifier[13,17].

To investigate this possibility we analysed stimulus-specific patterns of univariate activity in each of the decoding clusters (see Fig. 4). Using a *t*-test comparing activity for observed index and little finger movements, we classified each voxel in a binary fashion according to its preferred stimulus ($t > 0 =$ index-preferred, $t < 0 =$ little-preferred). Analysis of the univariate signal (beta values) revealed a significant interaction between congruency and stimulus preference ($F_{1,19} = 9.306$, $p = .007$, $\eta_p^2 = .329$) that did not vary across cluster ($F_{2,38} = 1.868$, $p = 0.168$, $\eta_p^2 = 0.089$). This interaction was driven by weaker activity on congruent relative to incongruent trials in voxels tuned away from the current stimulus ($t_{19} = 2.214$, $p = 0.039$), with no congruency effect in voxels tuned towards it ($t_{19} = 1.099$, $p = 0.286$). Therefore, in line with a sharpening account and inconsistent with cancellation models, this pattern suggests that suppression of sensory activity is found in voxels that are tuned away from, rather than tuned to, currently expected action outcomes.

## Discussion

We tested for the first time whether prediction during action operates in line with normative Bayesian models of perceptual processing, where it is widely assumed that predictive signals 'sharpen' expected sensory representations. Precisely as predicted under this account, we demonstrated that visual events congruent with concurrently performed actions were more readily decoded from visual brain activity than incongruent events. Furthermore, expectations were found to lead to suppressed activity in voxels tuned away from the expected stimulus. These effects of expectation generated on the basis of action are consistent with the assumption that predictive message-passing is a ubiquitous feature of cortical function[21].

Sharpening models propose that observers use their prior knowledge about the likelihood of different events to optimise perception in an uncertain world—biasing perceptual processing towards expected events, which in turn aids the rapid construction of (on average) more veridical percepts[12]. This sharpening is considered to arise through competitive interactions between neural populations tuned towards and away from the expected stimulus, such that activity in unpredicted units is attenuated relative to that in predicted units (e.g., through lateral inhibition)[13,17]. Predictive signals thereby stop 'gossiping' among sensory units[21] (for further discussion see ref.[22]). The competitive interaction process is likely started by pre-activating populations tuned towards expected features before a stimulus has been presented, given findings that early visual regions represent a stimulus in anticipation of its presentation[23]. Such a pre-activation mechanism concords with classic cognitive models of perception-action interactions, which assume that preparation of an action requires activating representations of its expected consequences[2].

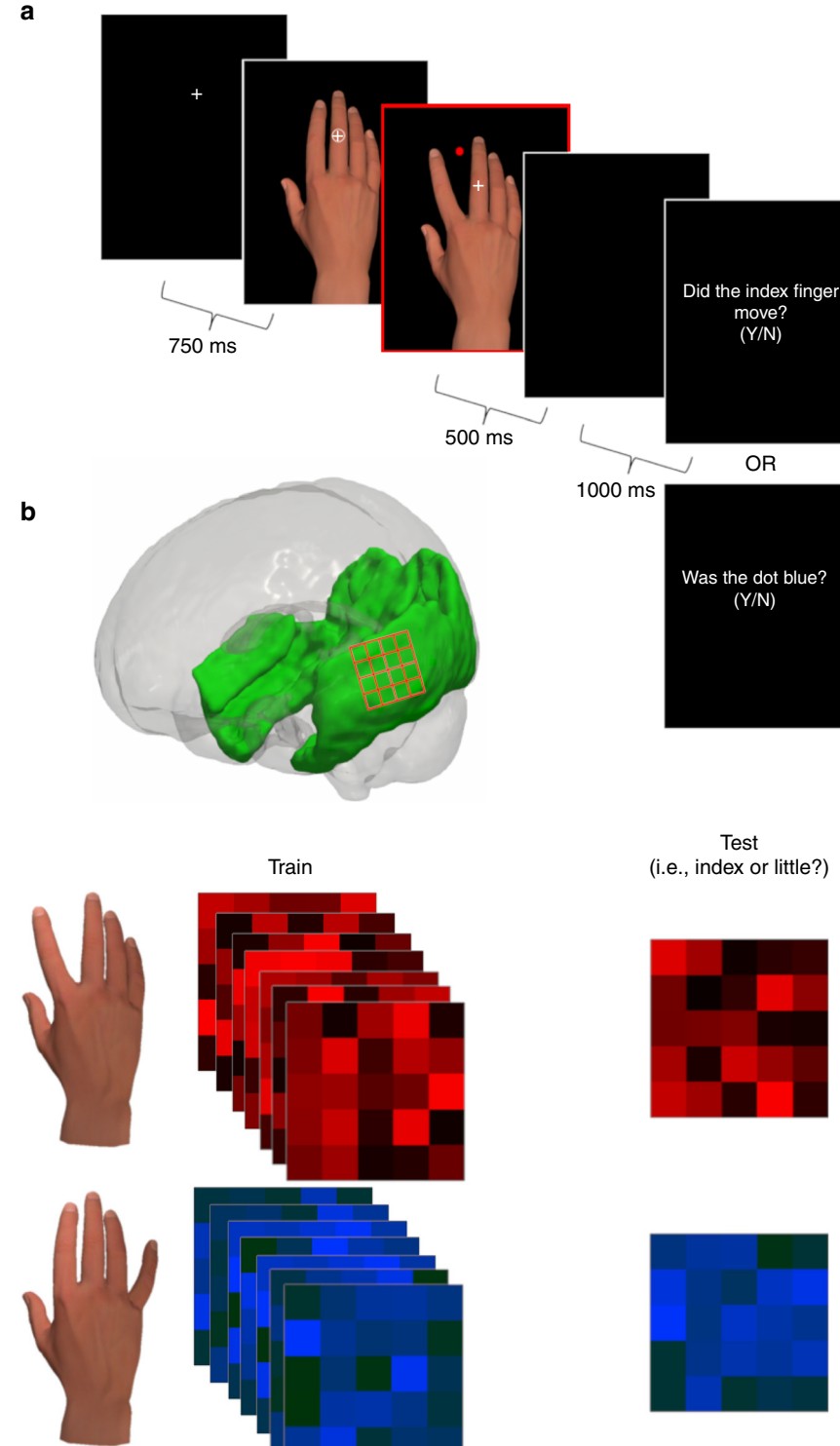

**Fig. 2** Action execution and observation task, with an illustration of pattern classification. **a** During the experimental task participants performed actions (e.g., index finger movement), which triggered movements of the avatar hand that were congruent (index finger) or incongruent (little finger) with their actions and in synchrony with them. **b** A searchlight approach[18] was used to decode the identity of observed actions based on BOLD activity from occipital and temporal regions (shaded in green)

Such a mechanism explains why predicted events are more readily detected[24], faster to enter conscious awareness[25] and appear phenomenally more intense[26]. The evidence presented here finds that prediction sharpens sensory processing of action outcomes in a similar fashion, suggesting that the same perceptual benefits may also be enjoyed in active settings. Such sharpening during action is likely to be adaptive given that we are also required to make sense of action outcomes in a noisy and rapidly changing sensory world. For example, if attempting to drink a cup of coffee in a dark kitchen before sunrise, we will generate more veridical percepts of our ongoing actions if we increase the weight on expected sensory channels (e.g., the sight

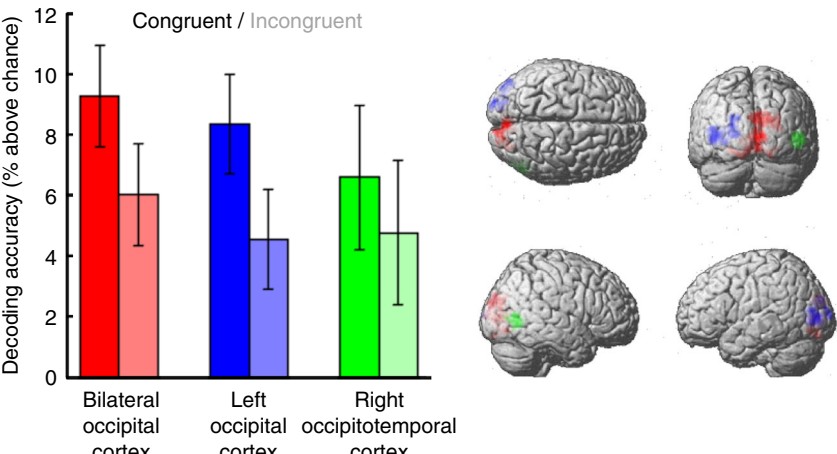

**Fig. 3** Stimulus decoding accuracies across regions of interest in the visual brain. Accuracies are shown for ROIs in bilateral occipital cortex (red), left occipital cortex (blue) and right occipitotemporal cortex (green). Decoding accuracy is higher on congruent (saturated) relative to incongruent (desaturated) trials, suggesting higher fidelity sensory representations when stimuli reflect expected action outcomes. Error bars show 95% within-participant confidence intervals of the mean difference between conditions (N=20, 95% CI/√2)[34]

of a moving hand). That percepts are generated rapidly is likely to be of paramount importance in action settings, given that the dynamic process of acting necessarily alters incoming sensory input on a millisecond timescale.

These findings are hard to reconcile with cancellation models from the action control literature, which suggest that activity in expected sensory units is suppressed during action. These models have implicitly assumed that prediction influences sensory processing differently when predictions are based upon one's own actions. They emphasise how it is adaptive for agents to ignore predictable sensations to remain maximally sensitive to unexpected outcomes that are more likely to be important for learning or planning new actions[3,5]. Previous reports supporting these models have inferred cancellation based on an attenuated univariate signal for predictable action outcomes. Interestingly we find that univariate visual activity is lower when observed events are congruent with action, as would be predicted under cancellation models, but only in voxels tuned away from currently expected action outcomes. It therefore appears likely that previously observed attenuations in visual[9–11] and somatosensory[5,8] areas reflect a dampening of responses in units tuned to unexpected stimuli, rather than the typically-assumed suppression in expected sensory units, and hence may not reflect the operation of a cancellation mechanism.

The cancellation model has retained a wide influence on research in cognitive neuropsychiatry, particularly when accounting for unusual beliefs about action and agency that arise in clinical conditions like schizophrenia. For instance, the fact that patients with delusions do not show neural signatures of cancellation when tickling themselves[27] has supported the idea that delusions about action (e.g., that your movements are controlled by an alien force) arise due to failures to suppress the sensory consequences of action, making movements appear like those that are externally generated[7]. If prediction during action instead sharpens sensory processing, the mechanism underlying these delusions may need revisiting. For example, if predictions ordinarily sharpen perception, deficits in prediction mechanisms may lead to heightened uncertainty about the perceptual consequences of our actions. This uncertainty may make us prone to developing unusual beliefs about our movements and their causes. Such an account would be concordant with broader computational models from neuropsychiatry which suggest that delusions about a wide variety of phenomena arise because weak

sensory evidence is given undue weight when making inferences about ourselves and the world around us[28].

As well as adjudicating between cancellation and sharpening models, this study presents a novel investigation of the relationship between expectations evoked by action and top-down attention, manipulated by task demands. Manipulations of expectation (what is likely to occur) are often confounded with top-down attentional demands (what is relevant for task performance[29]), both in laboratory settings and natural environments. Orthogonally manipulating these factors revealed that predictive benefits on decoding performance in bilateral and left occipital cortices were independent of top-down attention, in line with previous studies of expectation in other contexts[17]. However, an interaction was found in our right occipitotemporal region, with a stronger congruent enhancement when stimuli were irrelevant to the participants' task. This finding is not explicitly predicted by either a sharpening or cancellation account, but it may suggest that biases induced by expectation are sometimes larger for unattended stimuli[30]. It is worth noting that these specific distinctions between top-down expectation and attention depend on defining these mechanisms based on contrasting kinds of top-down knowledge (i.e., probability and task relevance). Opting to define these processes differently (e.g., describing effects of task-irrelevant probabilities as 'attentional') may render attention and expectation empirically and conceptually indissociable. Nevertheless, the conclusive point that can be drawn from the present findings is that sensory expectations sharpen sensory processing largely independently of task relevance.

In conclusion, the data presented here have shown that prediction during action sharpens sensory representations—visual events congruent with concurrently performed action were more readily decoded from visual brain activity, and expectations led to suppressed activity in voxels tuned away from the expected stimulus. Therefore, inconsistent with dominant action models, these results suggest that sensory processing during action is optimised in line with normative models of Bayesian perceptual inference, facilitating veridical perception of action outcomes in an inherently uncertain sensory world.

## Methods
**Participants**. Twenty healthy human volunteers (13 female, 7 male; age range 19–28 years, mean age = 23.1 years) participated in the study. This sample size was chosen to match a previous study that used the same multivariate decoding

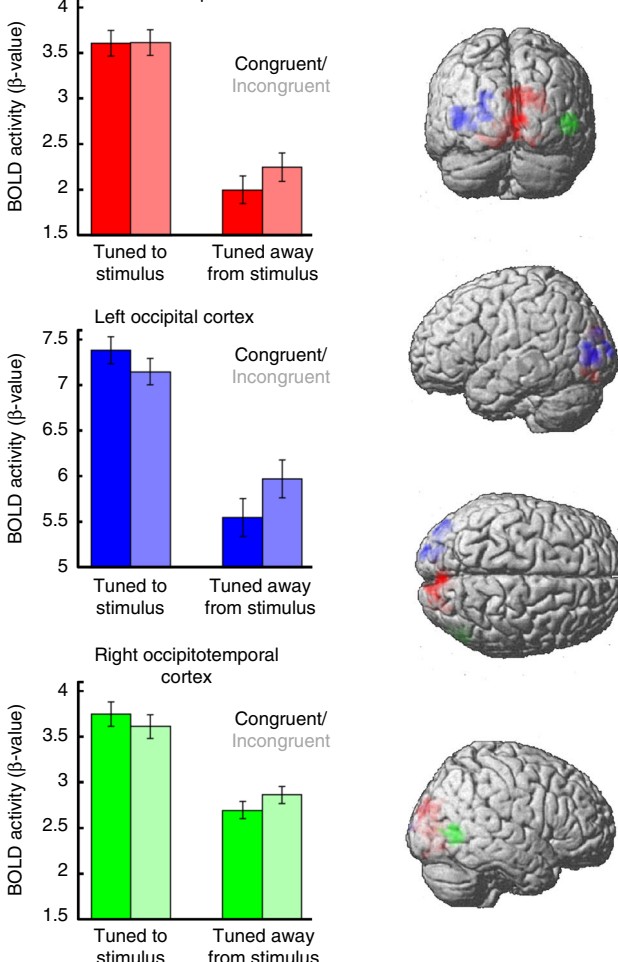

**Fig. 4** BOLD activity across regions of interest in the visual brain. Activity is lower for congruent (saturated) relative to incongruent (unsaturated) trials only in voxels that are tuned away from the observed stimulus (e.g., those voxels tuned towards index fingers on trials where little finger movements are presented). This pattern suggests that the lower BOLD signal found throughout the action literature when perceiving expected action outcomes may reflect a sharpening of expected sensory representations, rather than cancellation. Error bars show 95% within-participant confidence intervals of the mean difference between conditions ($N = 20$, 95% $CI/\sqrt{2}$)[34]

methodology in the context of a purely perceptual task and obtained robust effects of expectation[17]. An additional three participants were tested but data were not included in the final sample either due to excessive movement during scanning (one), premature termination of the experiment (one), or technical issues during scanning (one). All participants had normal or corrected-to-normal vision, and reported no current neurological or psychiatric illness. All participants provided written informed consent prior to participation and were reimbursed £10/h. All experimental procedures were reviewed and approved by the Birkbeck, University of London and University College London Ethics Committees.

**Procedure**. Stimuli were displayed against a black background on a rear-projection screen using a JVC DLA-SX21 projector ($26 \times 19.5$ cm, 60 Hz). Observed hand stimuli were generated in Poser 10 (Smith Micro Software) and consisted of a gender-neutral right hand viewed from a canonical first-person perspective (height ~ 13 degrees, width ~ 9 degrees, see Fig. 2). Participants lay supine in an MRI scanner with both hands placed on MR-compatible button boxes. The right-hand box was positioned across the midline of the participant's body, such that the index finger was above the little finger on the dorsal-ventral axis. Participants depressed two buttons on the right-hand box with their index and little finger except when executing movements. The left-hand button box was positioned below the right-hand box on the participant's left leg, and participants placed their left thumb between two response keys.

Each trial began with the presentation of a white fixation cross, which remained present throughout stimulus presentation. After 750 ms, a neutral hand image was presented behind the fixation cross. On congruent and incongruent trials, this neutral hand image was accompanied by a white shape (square or circle) indicating which action (index or little abduction) the participant was required to perform. The display remained on screen until participants executed the cued action, measured by the release of the depressed key on the button box. Upon release of the key, the neutral hand image was immediately replaced by an image of the avatar hand abducting either its index or little finger. This sequence created apparent motion of the observed finger that could be congruent or incongruent with the participant's own action, and that always appeared in synchrony with it. Congruency therefore reflects relative expectation – congruent action outcomes are more expected than incongruent ones, based either on inherited evolutionary expectations or a prior lifetime of learning about what is likely[2,5,8] (note that all statements are therefore relative throughout the manuscript, and a congruent suppression is equivalent to an incongruent facilitation). The movement of the avatar hand also revealed a coloured dot (red or blue) in the previous fingertip location (see Fig.1). On no move trials, an imperative shape cue did not appear with the neutral hand stimulus and the apparent motion sequence occurred after a fixed delay of 438 ms—matched to the average action execution reaction time in a pilot experiment. A fixed delay was implemented such that the onset of movement had approximately comparable temporal predictability relative to the trials where stimulus onset was yoked to the participant's action. On all trials, the hand image was removed after 500 ms and the screen was blanked for 1000 ms.

Participants completed one of two tasks, either making judgements about the identity of the observed finger abduction (e.g., 'Did the INDEX finger move?'—finger-judgement trials) or the colour of the dot revealed by the finger movement (e.g., 'Was the dot BLUE?'—colour judgement trials). On each trial the question was presented for 1500 ms, within which time participants were required to indicate their response via a keypress with their left thumb. The next trial began after a jittered inter-trial interval of 2–6 s.

The experiment was conducted in eight scanning sessions. Each session comprised 48 trials. On two-thirds of these trials participants executed index or little finger abductions with equal probability, and subsequently observed either congruent or incongruent action outcomes with equal probability (16 each). The remaining third of trials were *no move* trials (16), where participants observed index or little abductions without moving themselves. The task was blocked within each scanning session, such that one half of the session comprised finger-judgement trials and the other colour judgement trials. The task alternated across sessions, with the order counterbalanced across participants. At the beginning of each block, participants were reminded of the task they were performing, as well as the mapping between imperative shape cues and executed actions. This mapping was counterbalanced across participants, and was also reversed halfway through the experiment (i.e., the beginning of the fifth scanning session) to remove any confound between action-outcome congruency and cue-outcome congruency over the experiment.

Before beginning the main experiment, participants completed two practice blocks of 48 trials in a room outside the scanner. This practice block contained identical ratios of each trial type.

**Behavioural performance analysis**. A congruency (congruent, incongruent) by task (finger-judgement, colour-judgement) ANOVA on participant accuracies in the tasks revealed that participants were significantly more accurate when making judgements about dot colour than finger identity ($F_{1,17} = 40.656$, $p < 0.001$, $\eta_p^2 = 0.705$). There was also an interaction between congruency and task ($F_{1,17} = 11.130$, $p = 0.004$, $\eta_p^2 = 0.396$), reflecting superior accuracy on congruent trials relative to incongruent trials when making judgements about finger stimuli ($t_{17} = 3.954$, $p = 0.001$), but no effect of congruency when making judgements about dot colour ($t_{17} = -0.944$, $p = 0.358$). This pattern resembles that obtained in previous studies in the sensory cognition literature, where expectations facilitate behavioural performance but only when they are task-relevant[17]. Due to a technical fault, choice data could not be recovered for two participants on > 40% of trials, who were excluded from the above analyses. Including the data available for these participants did not alter any statistical patterns observed.

**fMRI acquisition and preprocessing**. Images were acquired using a 3T Trio MRI scanner (Siemens, Erlangen, Germany) using a 32-channel head coil. Functional images were acquired using an echo planar imaging (EPI) sequence (ascending slice acquisition, TR = 3.36 s, TE1/TE2 = 30/30.25 ms, 48 slices, voxel resolution 3 mm isotropic). Structural images were acquired using a magnetisation-prepared rapid gradient-echo (MP-RAGE) sequence (voxel resolution: 1 mm isotropic).

Images were preprocessed in SPM12. The first six volumes of each participant's data in each scanning run were discarded to allow for T1 equilibration. All functional images were spatially realigned to the first image and temporally-realigned to the 24[th] (middle) slice. The participant's structural image was then coregistered to the mean functional scan and segmented to estimate forward and inverse deformation fields to transform data from participant's native space into Montreal Neurological Institute (MNI) space, and vice versa.

**Multivariate decoding analyses**. MVPA analyses were implemented using the TDT toolbox. In each analysis, a linear SVM was trained to discriminate which stimulus (index or little) was observed on a given trial from patterns of BOLD activity across voxels. The initial step in each analysis was the specification of a general linear model (GLM) in SPM12 including a separate regressor for each stimulus type (e.g., observed index movement) in each experimental condition (e.g., *congruent* trials) in each scanning run. All regressors were modelled to the onset of the observed stimulus, movement parameters were included as nuisance regressors, and all model regressors were convolved with the canonical haemo-dynamic response function. This GLM generated eight beta images (one for each scanning run) for each stimulus type (index or little) in each experimental condition that were used for subsequent decoding analyses.

Separate SVMs were trained and tested on the 16 beta images (eight index and little) in each experimental condition (congruent and incongruent), using a leave-one-out cross-validation procedure. For each decoding step 14 images from seven scanning runs were used to estimate a linear discriminant function separating index and little movements, which was then applied to the remaining two beta images to classify them as either index or little. This procedure resulted in eight decoding steps, where each step reserved beta images from one of the eight scanning runs for classifier testing. The SVM's accuracy was calculated as the proportion of correctly classified images across all decoding steps.

**Defining regions of interest**. All analyses used a 'searchlight' approach[18], which involved building a separate SVM for each voxel in the brain using the beta values falling within a searchlight radius of 3 voxels (9 mm), and assigning the SVM's accuracy to the voxel upon which the searchlight was centred. This procedure yielded decoding maps in participant's native space indicating each voxel's decoding accuracy relative to chance level (50%; i.e., decoding accuracy of 60% is treated as 10%). To allow comparison across participants, these decoding maps were normalised into MNI space using the forward deformation fields estimated in preprocessing and smoothed using a 4 mm FWHM Gaussian kernel in SPM12.

To maximise sensitivity, MVPA analyses were initially conducted collapsing across finger judgement and colour judgement trials. Searchlight analyses from no move trials were used to define regions of interest. Maps from each participant were normalised and smoothed (described above), and subjected to a one-sample *t*-test in SPM12, using cluster-wise inference to identify contiguous voxels where decoding accuracy was significantly above chance at the group level[19]. This involves identifying individual voxels that passed a 'height' threshold ($p < 0.001$ uncorrected) and an 'extent' threshold applied to contiguous voxels that pass the height threshold (FWE $p < 0.05$[31],). This combination of thresholds has been shown to control appropriately for false-positive rates[20]. We restricted this contrast to occipital and temporal areas using the SPM12 atlas, to limit analyses to regions putatively involved in different aspects of visual processing[32], and analyses were not constrained to clusters of a minimum size. This analysis revealed three clusters in bilateral occipital cortex (bOC, 1825 voxels), left occipital cortex (lOC, 703 voxels) and right occipitotemporal cortex (rOTC, 374 voxels, see Fig. 2). Note that these specific ROIs may not generalise beyond the participants that we scanned. This is because one-sample t-tests on decoding measures do not support population inference, given that below-chance decoding accuracies are not meaningful[33]. However, the analyses used below to test our hypotheses investigate the difference in decoding accuracy between two conditions, rather than comparing against chance performance and so do support population inference. It is also worth noting that similar findings were obtained if defining the ROIs according to a permutation test approach[33] (see Supplementary Note 1).

**Effects of expectation on stimulus decoding**. To investigate effects of expectation during action on decoding accuracy, we extracted and averaged the decoding accuracies within each cluster separately for congruent and incongruent trials. These mean accuracies were then subjected to a cluster (bOC, lOC, rOTC) by congruency (congruent, incongruent) ANOVA.

To investigate possible interactions between expectations during action and top-down attentional relevance[29], additional searchlights were conducted separately for each combination of congruency (congruent, incongruent) and task (finger judgement, colour judgement). This procedure was identical to that described above, though segregating stimulus events halved the number of stimulus events used to model beta images for decoding. Mean decoding accuracies for each participant were calculated for each condition in each cluster, and these values were analysed using a cluster (bOC, lOC, rOTC) by congruency (congruent, incongruent) by task (finger judgements, colour judgements) ANOVA.

**Effects of expectation on stimulus-specific activity**. We investigated how expectations during action change the profile of activity across sensory populations by examining how stimulus-specific patterns of univariate BOLD activity varied between *congruent* and *incongruent* trials, within the same regions of interest. Using the same unnormalised, unsmoothed images used for multivariate decoding, we conducted a t-contrast in SPM12 for each participant comparing activity for observed index finger stimuli and observed little finger stimuli across all conditions. This contrast yielded a t-map for each participant where positive and negative

values reflected a voxel's preference for either index or little finger stimuli, respectively.

After assigning a preferred stimulus to each voxel, we extracted univariate BOLD signal (beta values) from each voxel separately for congruent and incongruent trials as a function of whether the stimulus was the preferred or non-preferred stimulus for a given voxel. For example, if a voxel was classified as 'index preferring', the univariate signal on *congruent* trials where an index finger was presented was congruent-preferred, whereas signal on the same trials was congruent-non-preferred for voxels classified as 'little preferring'. Univariate BOLD signal was extracted from each voxel in each of the clusters used for decoding and analysed with a cluster (bOC, lOC, rOTC) by congruency (congruent, incongruent) by preference (preferred stimulus, non-preferred stimulus) ANOVA. Analyses examining univariate main effects of congruency are reported in the Supplementary Information (see Supplementary Note 2).

**Statistical information**. Regions of interest were identified using cluster-wise inference on group-level decoding maps, with a combined primary voxel threshold ($p < 0.001$ uncorrected) and cluster-defining threshold (FWE $p < 0.05$) that appropriately controls false-positive rates[20]. For alternative information prevalence analyses, see Supplementary Note 1 and Supplementary Fig. 1. All inferential statistics evaluating differences between experimental conditions used an alpha level of 0.05. Assumptions of parametric tests were met. All error bars show 95% within-participant confidence intervals of the mean difference between conditions[34].

## Data availability
All relevant data will be made available by the authors upon request.

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

## Acknowledgements

This work was funded by The Leverhulme Trust (RPG-2016-105). S.J.G. was funded by a Royal Society University Research Fellowship. F.dL. was supported by The Netherlands Organisation for Scientific Research (NWO Vidi grant), the James McDonnell Foundation (JSMF scholar award) and the EC Horizon 2020 Program (ERC starting grant 678286). We are grateful to colleagues at the Wellcome Trust Centre for Neuroimaging, UCL for technical support.

## Author contributions

All authors contributed to the design of the study. D.Y. collected the data, which was analysed and interpreted in conjunction with all other authors. D.Y. wrote the manuscript and all authors were involved in revisions. C.P. supervised this work.

## Additional information

**Competing interests:** The authors declare no competing interests.

