## [Transparent Peer Review File · Nature Communications]

Reviewers' comments:

Reviewer #1 (Remarks to the Author):

Yon et al. present an interesting study in which they compare different models for the perceptual consequences of perceived actions, using fMRI. Specifically, they investigate whether suppression of sensory activity due to predictable actions acts primarily on representations of the expected or the unexpected action, which can differentiate between sharpening models (which are popular in perception research) and established models of action control. The study is interesting, novel, and straightforward, mimicking a similar approach that has been taken in the domain of visual perception previously.

Major

The authors used a group level t-test to identify regions of interest with above chance classification performance. This approach has been called into question because classification performance cannot be below chance level. As laid out in Allefeld et al. (2016), the null hypothesis that the t-test approach examines is whether no subject shows classification accuracy, and hence rejecting it only says that some subjects showed above chance classification. It also does not generalize to the population. It is hence unlikely that the authors want to test this rather uninteresting hypothesis. An alternative is to use 'information prevalence', described in the abovementioned paper by Allefeld et al.

Both the standard and the sharpening account predict that expected stimuli should lead to a suppression of activity. However, I failed to find a contrast in the manuscript in which this straightforward prediction is actually tested. I think this should be added, because failing to find a general suppression effect calls both types of models into question. I was actually wondering why the authors did not use suppression of predicted relative to unpredicted conditions as a way to identify their regions of interest? In Figure 4, the authors show lower activity in voxels tuned away from the stimulus than in units tuned to the stimulus, and a modulation by congruency, but this does not show or test that there was an overall suppression. It is also unclear whether the activity on incongruent conditions is enhanced or the activity on congruent conditions is suppressed in the voxels tuned away, because only suppression would be predicted by the sharpening account.

Minor

- In the methods section, the authors first describe an approach in which they identify contiguous voxels without minimum cluster size constraints, and then describe a multiple comparison correction procedure with a minimum cluster size. It is currently not clear how these two approaches relate or why no correction for multiple comparisons was applied in the first step. Please clarify.

- The authors use a method described by Loftus to compute confidence intervals. In the paper by Loftus, multiple approaches to compute CIs for multi-level, within subjects designs are described. It is unclear which particular method was used in the current paper.

- Please report F/t-values and effect size measures also for all non-significant results.

- On page 7, the authors conclude that "prediction during action enhances the quality of sensory representations associated with expected outcomes". This conclusion seems premature at this stage and would only be warranted after the analysis that takes voxel tuning into account. For example, suppression of predicted in the absence of suppression of unpredicted information could also lead to this result. Please rephrase/remove.

- On page 9, the authors state that "this interaction was driven by weaker activity on congruent relative to incongruent trials in units tuned away from the current stimulus". Please replace "units"

by "voxels".

- The legend of Figure 3 is not very intuitive. Please consider a different legend or labeling the x-axis instead.
- Given that the authors did not perform a whole-brain analysis, it should be made clear in the figures which parts of the brain were actually investigated. A plane or shading could for example be used.
- I think it would make sense to briefly mention the behavioral results in the main text.

Reviewer #2 (Remarks to the Author):

The study 'Action sharpens sensory representations of expected outcomes' investigates decoding of observed behaviour from brain activation that is either congruent or incongruent to an executed action. Decoding accuracy is higher for congruent than for incongruent action. Furthermore, voxels that are tuned away from the stimulus are less activated in incongruent than in congruent trials. I think this is a very interesting study, addressing a crucial question in motor control. While I agree that the presented data are not consistent with a sensory suppression account, I think they are also consistent with a more general motor control model (see below). In my opinion, however, such an alternative interpretation does not compromise the important finding that the data are not consistent with the sensory suppression model.

I wonder whether the reported findings can be explained without referring to a sharpening account. The authors cite TEC theory, which assumes that actions are controlled by anticipating the sensory consequences of the action. If participants plan a specific behaviour, they will activate a sensory representation of this behaviour. In the congruent case, the same action representation is activated by motor planning and movement observation. In the case of an incongruent action, both motor planning and action observation activate different sensory representations. Such a model can explain the relatively reduced univariate activation in congruent trials compared to incongruent trials and the higher decoding in congruent trials. Furthermore, it is also compatible with the observation that activation in voxels tuned away from the stimulus is higher in incongruent compared to congruent trials. Given that the sharpening account is not directly tested but inferred from the data, I think the motor control account outlined above is more parsimonious.

Reviewer #3 (Remarks to the Author):

Yon and colleague report important and topical work. They use a simple action-based expectation manipulation to explore visual cortical responses to congruent (expected) or incongruent (unexpected) visual stimuli in order to distinguish between two related but partially-competing models. A standard model would suggest that action leads to ignoring or cancelling an expected outcome while the alternative Bayesian model would predict an expectation-based sharpening of the representation of that outcome. While both might predict an overall reduction in neural response as measured using standard fMRI, MVPA analysis was used to distinguish these two possibilities, providing support for the latter model. Notably, the accuracy of the trained decoder was greater for congruent than incongruent trials suggestive of the hypothesised "sharpening" effect. Complementary univariate analysis showed that activity in units tuned away from the congruent outcome was suppressed.

I enjoyed and admired this paper. It is clearly presented and well-motivated as well as elegant in

design and analysis. I think that it will add to and challenge the existing literature and will have important implications for computational psychiatry. I would like, if possible, for the authors to perhaps think a little more in relation to the conflict between cancellation/sharpening models. This does seem to be something of a confused area within predictive coding models which variously seem to assume that tope down expectation does sharpen representation or alternatively enhanced prediction error signals with only the latter being fed forward. Is it possible that both effects go hand in hand? Or do the current results (from the complementary univariate analysis) seem to rule out the latter possibility? I realise that the authors don't have much room to speculate and they already make some useful comments in regard to the cancellation models so they may not wish to act on this comment.

The other question I would like to raise – and perhaps one that many readers of the paper will consider, even if only fleetingly – is whether we may be looking at a primarily attentional effect – i.e. does moving a particular finger divert attention to the visual representation of that finger and could this account for both behavioural and neural findings?

We are very grateful to you and the reviewers for your enthusiasm and helpful comments. We have improved the manuscript on the basis of these suggestions. Most notably, we have conducted the additional analyses suggested by Reviewer 1 and incorporated these into our manuscript. We highlight the changes to the manuscript in red.

Reviewers' comments:

Reviewer #1 (Remarks to the Author):

Yon et al. present an interesting study in which they compare different models for the perceptual consequences of perceived actions, using fMRI. Specifically, they investigate whether suppression of sensory activity due to predictable actions acts primarily on representations of the expected or the unexpected action, which can differentiate between sharpening models (which are popular in perception research) and established models of action control. The study is interesting, novel, and straightforward, mimicking a similar approach that has been taken in the domain of visual perception previously.

Major

The authors used a group level t-test to identify regions of interest with above chance classification performance. This approach has been called into question because classification performance cannot be below chance level. As laid out in Allefeld *et al.* (2016), the null hypothesis that the t-test approach examines is whether no subject shows classification accuracy, and hence rejecting it only says that some subjects showed above chance classification. It also does not generalize to the population. It is hence unlikely that the authors want to test this rather uninteresting hypothesis. An alternative is to use ‘information prevalence’, described in the abovementioned paper by Allefeld *et al.*

We agree with the reviewer that the t-test investigating above-chance classification performance is not suitable for population inference, and relatedly, may only reflect effects in some of the participants. However, we did not use this approach directly to test any hypotheses, only to identify candidate regions of interest (ROIs) that may in principle be involved in processing the stimuli used in this experiment in the participants we tested. Given this aim, our ROI-defining analysis might be seen as conceptually analogous to a functional localiser, limiting analyses to voxels that may be involved in a domain of interest. The logic holds even if the resultant ROIs are defined based upon effects present only in a handful of participants. Crucially, our hypothesis-testing contrast was a comparison between decoding accuracies in two conditions. This contrast can meaningfully take values both above and below zero (i.e. congruent > incongruent, or incongruent > congruent), so the argument made by Allefeld *et al.*¹ does not apply to this analysis. Therefore, although we fully agree with the reviewer that our statistical approach does not allow the precise ROIs to be generalised to the wider population, we do not believe that the logic of our approach requires this, in much the same way as studies using a functional localiser are not invalidated by the fact that the ROIs are specific to the participants tested. The key point is that the analysis of congruency that we use to distinguish sharpening versus cancellation (suppression) models is conducted in a manner that does allow population inference. We also note that the t-test approach continues to be standard in the field, even in very recent publications by the same group who identified this potential problem (e.g.²⁻⁴).

By retaining the t-test approach we hence believe that we do not compromise the logic of our statistical argument. We also use a more standard methodology that would be understood by a wider number of readers and is consistent with the method used for our hypothesis-testing analyses. Therefore, in our revised manuscript we now discuss the theoretical arguments made by Allefeld *et al.* on p. 20 and draw a clear distinction between our ROI-defining analysis (which does not support population inference) and our hypothesis-testing analyses (which do).

Nevertheless, we wished to find out whether our main conclusions could be corroborated with the permutation test approach suggested by Allefeld *et al.* We purchased more memory for the computer and investigated information prevalence by permuting class labels used for multivariate decoding (128 unique permutations), using the resultant decoding maps to create second-level permutations (1,000,000 random permutations) that were compared to our real, unpermuted data (see ¹ for further details). First, as a ‘sanity check’ we investigated information prevalence in the original ROIs used in our analyses. The prevalence of information about the observed stimuli was high and highly significant (estimated prevalence = .721, rejecting the majority null at $p=.003$). However, we acknowledge that this is a circular analysis, given that the original ROIs were defined based on above-chance decoding. Next, we used the information prevalence approach to define voxels from which information about the observed stimuli was significantly prevalent (i.e. above chance in a majority of participants). In these voxels, decoding accuracy was higher for the congruent than incongruent condition ($p=.025$), replicating the main analysis supporting the sharpening over the cancellation model. Finally, these voxels also showed a congruency x stimulus preference interaction in the univariate signal ($p=.012$). Therefore, even

using the approach suggested by Allefeld *et al.* we still find the patterns of results that support the sharpening over the cancellation model.

For the reasons outlined above, our preference is to continue using the standard t-test approach in our manuscript but we have added to the manuscript that a similar pattern was obtained if defining the ROIs according to a permutation test approach (p.20), and given the details of these analyses in Supplementary Materials.

Both the standard and the sharpening account predict that expected stimuli should lead to a suppression of activity. However, I failed to find a contrast in the manuscript in which this straightforward prediction is actually tested. I think this should be added, because failing to find a general suppression effect calls both types of models into question. I was actually wondering why the authors did not use suppression of predicted relative to unpredicted conditions as a way to identify their regions of interest? In Figure 4, the authors show lower activity in voxels tuned away from the stimulus than in units tuned to the stimulus, and a modulation by congruency, but this does not show or test that there was an overall suppression. It is also unclear whether the activity on incongruent conditions is enhanced or the activity on congruent conditions is suppressed in the voxels tuned away, because only suppression would be predicted by the sharpening account.

We agree that a range of studies have shown an overall suppression of activity for expected stimuli, and that it was partly these findings that inspired the cancellation model and subsequently a sharpening account of the same data patterns. However, while the models can be true given the patterns in data, it is not the case that the models necessarily predict these patterns for all sensory regions. As outlined in the text and shown in Fig. 1, both cancellation and sharpening models predict that reductions in activity should not occur uniformly over sensory populations, but should occur primarily in voxels tuned to expected (cancellation) or unexpected (sharpening) stimuli. Therefore, the models predict a reduction in activity either in voxels tuned towards expected or unexpected stimuli, but not both. Given that we did indeed find a reduction in activity for one of these two classes of voxels (those tuned towards unexpected stimuli), our results are in line with this prediction. However, the models do not necessarily predict a reduction in activity when collapsing across all voxels. In particular, the sharpening model may predict greater activity in expected trials in voxels tuned to expected stimuli, generating effects that oppose the suppression in voxels tuned away from the stimulus. For instance, de Lange *et al.*⁵ state ‘that [units] with the highest selectivity may even be enhanced’ (p.7).

This logic means that for sharpening models specifically to predict a global expectation suppression effect, may require the assumption that the regions in question contain a strong predominance of units tuned to unexpected events rather than expected events. It is worth noting that overall suppression effects are not a ubiquitous feature of expectation (e.g.,⁶), perhaps for this reason. Our study differs from others looking at expectation by defining the ROIs according to which voxels contain information about our two presented stimulus types (index and little fingers), given that pattern classification and stimulus-selectivity analyses represented our primary aim. Defining ROIs in this fashion will bias toward selection of voxels tuned to stimuli presented in this experiment, and away from voxels tuned to other types of stimuli. For this reason, our ROIs likely contain relatively more units tuned to expected events compared to other approaches, and likely for this reason we do not find an overall univariate (significant) suppression effect in these ROIs (p=.439). Previous studies have often defined ROIs using functional localisers, but we did not incorporate subject-specific functional localisers given that we did not deem this appropriate for our primary aim. However, at the coordinates where previous overall suppression effects have been found in studies with a similar design we find weak

evidence in line with these previous findings despite differences in participants and the specific stimuli used ($t_{19} = 1.752, p < .05$, one-tailed).

We have reworded our introduction to make clearer that the models do not necessarily predict an overall suppression effect for predicted stimuli and how our novel stimulus-specific approach allows the divergent predictions of these models to be tested (p. 3-4). We have added the statistics for the overall univariate effects both to the Supplementary Materials, and referenced these in the main text.

The Reviewer also raises an interesting point concerning whether effects reflect suppression of congruent activity or enhancement of incongruent activity in voxels tuned away from the current stimulus. By definition, there is lower activation in voxels tuned away from the presented stimulus, rather than towards it. Unfortunately it is therefore not possible to distinguish these accounts because the effects of interest are determined by comparing neural patterns on expected and unexpected trials, and suppression of activation in one condition would look identical to enhancement in the other (note that *no move* trials are qualitatively different from *congruent* and *incongruent* trials – due to performance of a single rather than dual task – and the sensory event is arguably unexpected on these trials in any case, given that all sensations are either expected or unexpected). Arguably this empirical quandary is mirrored by a theoretical one – since an event is always expected or unexpected, and it is unclear what could constitute a theoretically appropriate baseline. We now discuss explicitly in the Methods section that all comparisons are relative and therefore that suppression of congruent activity is equivalent to enhancement of incongruent activity (p.16).

Minor

- In the methods section, the authors first describe an approach in which they identify contiguous voxels without minimum cluster size constraints, and then describe a multiple comparison correction procedure with a minimum cluster size. It is currently not clear how these two approaches relate or why no correction for multiple comparisons was applied in the first step. Please clarify.

We used a common form of cluster-based inference⁷ implemented in SPM where significant clusters are identified by setting a threshold for the size of an effect in each voxel within the cluster (the height threshold – $p < .001$ uncorrected), and a threshold for the number of contiguous voxels surviving this height threshold (the extent threshold - FWE $p < .05$). This combination of thresholds has been shown to control appropriately for false positive rates⁸.

Our reference to having ‘no minimum cluster size’ was intended to make clear that we did not use an additional constraint on the size of these clusters (e.g. some recommend only analysing clusters with a minimum extent of 10 voxels⁹). We have reworded the relevant parts of the manuscript to make this point clearer (p.19).

- The authors use a method described by Loftus to compute confidence intervals. In the paper by Loftus, multiple approaches to compute CIs for multi-level, within subjects designs are described. It is unclear which particular method was used in the current paper.

We have clarified in all relevant figure legends that error bars display confidence intervals on the difference between means (p.8 and p.10).

- Please report F/t-values and effect size measures also for all non-significant results.

These are now reported.

- On page 7, the authors conclude that “prediction during action enhances the quality of sensory representations associated with expected outcomes”. This conclusion seems premature at this stage and would only be warranted after the analysis that takes voxel tuning into account. For example, suppression of predicted in the absence of suppression of unpredicted information could also lead to this result. Please rephrase/remove.

We agree and have rephrased “this result suggests...” to “this result is consistent with the idea that..”(p.8).

- On page 9, the authors state that “this interaction was driven by weaker activity on congruent relative to incongruent trials in units tuned away from the current stimulus”. Please replace “units” by “voxels”.

We have made the suggested replacement (p.9).

- The legend of Figure 3 is not very intuitive. Please consider a different legend or labeling the x-axis instead.

We have clarified the legend on Figure 3 (p.8).

- Given that the authors did not perform a whole-brain analysis, it should be made clear in the figures which parts of the brain were actually investigated. A plane or shading could for example be used.

We have added relevant shading to the schematic illustration of our searchlight analysis in Figure 2, and explained what it illustrates in the figure legend (p.7).

- I think it would make sense to briefly mention the behavioral results in the main text.

We have now mentioned these findings briefly in the way described (p.6).

Reviewer #2 (Remarks to the Author):

The study ‘Action sharpens sensory representations of expected outcomes’ investigates decoding of observed behaviour from brain activation that is either congruent or incongruent to an executed action. Decoding accuracy is higher for congruent than for incongruent action. Furthermore, voxels that are tuned away from the stimulus are less activated in incongruent than in congruent trials. I think this is a very interesting study, addressing a crucial question in motor control. While I agree that the presented data are not consistent with a sensory suppression account, I think they are also consistent with a more general motor control model (see below). In my opinion, however, such an alternative interpretation does not compromise the important finding that the data are not consistent with the sensory suppression model.

I wonder whether the reported findings can be explained without referring to a sharpening account. The authors cite TEC theory, which assumes that actions are controlled by anticipating the sensory consequences of the action. If participants plan a specific behaviour, they will activate a sensory representation of this behaviour. In the congruent case, the same action representation is activated by motor planning and movement observation. In the case of an incongruent action, both motor planning and action observation activate different sensory representations. Such a model can explain the relatively reduced univariate activation in congruent trials compared to incongruent trials and the higher

decoding in congruent trials. Furthermore, it is also compatible with the observation that activation in voxels tuned away from the stimulus is higher in incongruent compared to congruent trials. Given that the sharpening account is not directly tested but inferred from the data, I think the motor control account outlined above is more parsimonious.

We agree that our findings are consistent with TEC, and indeed that TEC and sharpening accounts are in present forms compatible at a wider level but with a different focus. Notably, sharpening accounts aim to explain neural processing of expectations while TEC is focussed on effects of action on perception (and vice versa). It is stated in the exposition of TEC¹⁰ that ‘As regards perception, its focus is on “late” cognitive products of perceptual processing that stand for, or represent, certain features of actual events in the environment. TEC does not consider the complex machinery of the “early” sensory processes that lead to them.’ (p.849) While TEC is not primarily concerned with neural mechanisms, it would likely consider that preactivation of expected units is responsible for the sharpening observed in the present dataset. What we had not made clear in the original version of the manuscript is that the sharpening model also predicts such preactivation. The sharpening model may additionally predict that lateral inhibition between neighbouring units contributes to these effects. While this is not an explicit prediction of TEC to our knowledge, it is likely that this idea is consistent with TEC and notable that such lateral inhibition is a frequently observed property of the visual system¹¹ and therefore likely requires accommodation.

Given that the sharpening model relates specifically to neural processing, we have continued to frame our findings in these terms. However, we now outline explicitly in the Discussion that such sharpening is thought to arise from preactivation of expected units, and that this represents a core prediction also of TEC (p.11). Furthermore, we have made clearer throughout the manuscript that any claims about levels of activation associated with congruent stimuli are relative to incongruent stimuli, and that it is empirically and theoretically difficult to establish absolute baselines (see response to Reviewer 1).

Reviewer #3 (Remarks to the Author):

Yon and colleague report important and topical work. They use a simple action-based expectation manipulation to explore visual cortical responses to congruent (expected) or incongruent (unexpected) visual stimuli in order to distinguish between two related but partially-competing models. A standard model would suggest that action leads to ignoring or cancelling an expected outcome while the alternative Bayesian model would predict an expectation-based sharpening of the representation of that outcome. While both might predict an overall reduction in neural response as measured using standard fMRI, MVPA analysis was used to distinguish these two possibilities, providing support for the latter model. Notably, the accuracy of the trained decoder was greater for congruent than incongruent trials suggestive of the hypothesised “sharpening” effect. Complementary univariate analysis showed that activity in units tuned away from the congruent outcome was suppressed.

I enjoyed and admired this paper. It is clearly presented and well-motivated as well as elegant in design and analysis. I think that it will add to and challenge the existing literature and will have important implications for computational psychiatry. I would like, if possible, for the authors to perhaps think a little more in relation to the conflict between cancellation/sharpening models. This does seem to be something of a confused area within predictive coding models which variously seem to assume that tope down expectation does sharpen representation or alternatively enhanced prediction error signals with only the latter being fed forward. Is it possible that both effects go hand in hand? Or do the current

results (from the complementary univariate analysis) seem to rule out the latter possibility? I realise that the authors don't have much room to speculate and they already make some useful comments in regard to the cancellation models so they may not wish to act on this comment.

We agree with the reviewer that predictive coding models make ambiguous predictions here. Importantly for our study, the specific cancellation and sharpening models do not contain these ambiguities and make mutually exclusive predictions about the patterns of data we should obtain. The general predictive coding scheme suggests that there are distinct populations of representation units and prediction error units, and allow for the possibility that cancellation-like effects occur only within the error unit population. However, these models do not specify the relative proportion of representation and error units, meaning it is not possible to derive a clear prediction as to whether sharpening or cancellation should be observed in the BOLD signal of our study. It is therefore difficult to comment on the consistency between the current results and the predictive coding framework. Given this ambiguity, we have opted not to discuss this framework at length, though we now reference the predictive coding scheme in our Discussion section, alongside a recent paper¹² that further discusses ambiguities between cancellation and sharpening mechanisms in this model (p. 11).

The other question I would like to raise – and perhaps one that many readers of the paper will consider, even if only fleetingly – is whether we may be looking at a primarily attentional effect – i.e. does moving a particular finger divert attention to the visual representation of that finger and could this account for both behavioural and neural findings?

We shared the reviewers' interest in the extent to which any expectation effects would reflect attention. It was for this reason that we included the factor of task relevance of stimuli, given that researchers often distinguish predictive and attentional effects as those driven by stimulus probability and task-relevance¹³, respectively. By this definition our neural effects of congruency are not attentional in at least two of our three ROIs as they are obtained irrespective of task-relevance (note that the expectation effect interacted with task-relevance in the right occipitotemporal ROI such that expectation effects are strongest when task *irrelevant*, making attentional explanations of the expectation effect itself also unlikely in this ROI). It is additionally worth noting that the effects we report do not mirror those obtained in previous studies of top-down attention. For example, neuroimaging experiments tend to find that when participants are cued to attend to a stimulus this leads to robust enhancements in the activity of voxels tuned to the attended stimulus, but has no effect on voxels that are tuned to the unattended stimuli¹⁴. Our univariate analysis finds the opposite pattern – that action-outcome congruency has its effects on voxels that are tuned away from the expected stimulus, which is also what is found for predictable stimuli when attentional focus is orthogonally manipulated¹⁵.

Nevertheless, we also acknowledge that there are a range of definitions of attention, and a range of levels of description of mechanism. Opting to define processes differently (e.g. describing effect of task-irrelevant probabilities as 'attentional') may render attention and expectation empirically and conceptually indistinguishable. We now outline in the Discussion that claims of sensory sharpening being independent of attention will only hold for the present definitions (pp. 13-14).

Response to Reviewers References

1. Allefeld, C., Görden, K. & Haynes, J.-D. Valid population inference for information-based imaging: From the second-level t-test to prevalence inference. *NeuroImage* **141**, 378–392 (2016).
2. Christophel, T. B., Iamshchinina, P., Yan, C., Allefeld, C. & Haynes, J.-D. Cortical specialization for attended versus unattended working memory. *Nat. Neurosci.* **21**, 494–496 (2018).
3. Loose, L. S., Wisniewski, D., Rusconi, M., Goschke, T. & Haynes, J.-D. Switch-independent task representations in frontal and parietal cortex. *J. Neurosci. Off. J. Soc. Neurosci.* **37**, 8033–8042 (2017).
4. Pischedda, D., Görden, K., Haynes, J.-D. & Reverberi, C. Neural representations of hierarchical rule sets: the human control system represents rules irrespective of the hierarchical level they belong to. *J. Neurosci.* 3088–16 (2017). doi:10.1523/JNEUROSCI.3088-16.2017
5. Lange, F. P. de, Heilbron, M. & Kok, P. How do expectations shape perception? *Trends Cogn. Sci.* **0**, (2018).
6. Turk-Browne, N. B., Scholl, B. J., Chun, M. M. & Johnson, M. K. Neural evidence of statistical learning: efficient detection of visual regularities without awareness. *J. Cogn. Neurosci.* **21**, 1934–1945 (2009).
7. Friston, K. J., Holmes, A., Poline, J. B., Price, C. J. & Frith, C. D. Detecting activations in PET and fMRI: levels of inference and power. *NeuroImage* **4**, 223–235 (1996).
8. Eklund, A., Nichols, T. E. & Knutsson, H. Cluster failure: Why fMRI inferences for spatial extent have inflated false-positive rates. *Proc. Natl. Acad. Sci. U. S. A.* **113**, 7900–7905 (2016).
9. Lieberman, M. D. & Cunningham, W. A. Type I and Type II error concerns in fMRI research: re-balancing the scale. *Soc. Cogn. Affect. Neurosci.* **4**, 423–428 (2009).
10. Hommel, B., Müsseler, J., Aschersleben, G. & Prinz, W. The theory of event coding (TEC): a framework for perception and action planning. *Behav. Brain Sci.* **24**, 849–878; discussion 878-937 (2001).
11. Angelucci, A., Levitt, J. B. & Lund, J. S. Chapter 29 Anatomical origins of the classical receptive field and modulatory surround field of single neurons in macaque visual cortical area V1. in *Progress in Brain Research* **136**, 373–388 (Elsevier, 2002).
12. Teufel, C., Dakin, S. C. & Fletcher, P. C. Prior object-knowledge sharpens properties of early visual feature-detectors. *Sci. Rep.* **8**, 10853 (2018).
13. Summerfield, C. & Egner, T. Feature-based attention and feature-based expectation. *Trends Cogn. Sci.* **20**, 401–404 (2016).
14. Serences, J. T., Saproo, S., Scolari, M., Ho, T. & Muftuler, L. T. Estimating the influence of attention on population codes in human visual cortex using voxel-based tuning functions. *NeuroImage* **44**, 223–231 (2009).
15. Kok, P., Jehee, J. F. M. & de Lange, F. P. Less is more: expectation sharpens representations in the primary visual cortex. *Neuron* **75**, 265–270 (2012).

REVIEWERS' COMMENTS:

Reviewer #1 (Remarks to the Author):

I would like to thank the authors for their revisions of the manuscript, which address the points I previously raised. My only remaining suggestions would be to remove the first step in the information prevalence analysis from the manuscript, because it is circular and hence does not add anything relevant. It would also be advisable to add a figure with the results of the information prevalence analysis to the supplemental material.

Reviewer #2 (Remarks to the Author):

The authors have addressed my main point sufficiently. I think this manuscript provides a novel and interesting contribution to the literature on predictive coding and the consequences of action on perception.

Reviewer #3 (Remarks to the Author):

The authors' responses to my questions are completely satisfactory to me